# Unsupervised Conditional Generation using noise engineered mode matching GAN

## Abstract

Conditional generation refers to the process of sampling from an unknown distribution conditioned on semantics of the data. This can be achieved by augmenting the generative model with the desired semantic labels, albeit it is not straightforward in an unsupervised setting where the semantic label of every data sample is unknown. In this paper, we address this issue by proposing a method that can generate samples conditioned on the properties of a latent distribution engineered in accordance with a certain data prior. In particular, a latent space inversion network is trained in tandem with a generative adversarial network such that the modal properties of the latent space distribution are induced in the data generating distribution. We demonstrate that our model, despite being fully unsupervised, is effective in learning meaningful representations through its mode matching property. We validate our method on multiple unsupervised tasks such as conditional generation, dataset attribute discovery and inference using three real world image datasets namely MNIST, CIFAR-10 and CelebA and show that the results are comparable to the state-of-the-art methods.

## 1 Introduction

Unsupervised learning or extraction of semantically relevant information from unlabelled data is a problem of great interest. It is useful for a wide range of tasks such as feature learning for supervised and semi-supervised classification, data augmentation and reconstruction (Hastie et al. (2009); Hinton et al. (2006); Hinton & Salakhutdinov (2006); Bengio et al. (2013); Yann (1987); Radford et al. (2015); Kingma et al. (2014); Cheung et al. (2014)). From a generative model's perspective, unsupervised learning amounts to divide the data manifold into semantically interpretable regions and sample new points from each of the regions (Jaakkola & Haussler (1999); Hastie & Tibshirani (1996)). However, distributions of most of the naturally occurring data such as images and speech lie over complex high-dimensional manifolds and do not have tractable forms.

Deep generative neural networks, in particular Generative Adversarial Networks (GANs) (Goodfellow et al. (2016)), are shown to be effective in modelling complex distributions for several real-world tasks (Radford et al. (2015); Isola et al. (2017); Van Den Oord et al. (2016); Odena et al. (2016); Denton et al. (2015)). Basic idea of a GAN is to implicitly learn sampling from the data generating distribution without explicitly estimating it. It is done by adopting an adversarial learning principle where the problem of sampling from an unknown data distribution is formulated as a min-max game between two function approximators called generator and discriminator. A generator learns to map an input noise distribution to the data generating distribution. This is facilitated by the discriminator which is trained to distinguish between the generated and the true samples. At equilibrium, the distribution of the generated samples converges to the data generating distribution. There are two major differences between GANs and conventional generative models - First, in GANs, the posterior of the latent variable conditioned on the data is intractable. This is due to the fact that GANs are not designed to explicitly model the relation between the latent and data space. If this relation can be inferred in a GAN setting, they can be readily used to perform a variety of unsupervised learning tasks. Second, in conventional generative models, the prior on the latent variable is chosen in accordance with some guess on the structure of the data. For example, number of clusters in K-means is chosen to be ten if there are ten 'classes' in the data (Jain et al. (1999)). As a result, the generated space inherits the properties of the latent space. Motivated by the these observations, in this paper, we propose a GAN model in which latent space, engineered in accordance with the data

space, is inverted back from the generated space. This enables the generation to be such that the modal properties of the data generating distribution are matched to those of the latent distribution.

The idea of inverting the latent space while training a GAN has been incorporated in different contexts (Chen et al. (2016); Donahue et al. (2016); Dumoulin et al. (2016); Mescheder et al. (2017); Larsen et al. (2015); Makhzani et al. (2015); Kim et al. (2017); Springenberg (2015); Bao et al. (2017); Sohn et al. (2015)). It is shown to stabilize the training and prevent a degenerative solution for the generator (known as mode collapse) (Che et al. (2016); Srivastava et al. (2017); Arjovsky et al. (2017); Salimans et al. (2016)). Although our work has a similar ambit, the following are our unique contributions:

- We propose to engineer the noise space according to the desired objective for unsupervised learning.
- We combine the latent space engineering with an inverse mapping from the data space to the latent space in order to enforce the matching the modal properties of generated data distribution with the engineered latent space.
- We include an activity regularizer on latent space reconstruction to avoid intra-mode collapse.
- Unlike some of the recent approaches (Chen et al. (2016); Srivastava et al. (2017)), which rely on minimizing a lower bound on the actual cost via variational inference, the loss terms in the proposed method are exact.

We demonstrate the use of the proposed method in several unsupervised learning tasks such as conditional generation, inference and attribute discovery through experiments on multiple real-world datasets[1].

## 2 PROPOSED METHOD

Let $P_X$ denote the data generating distribution from which the samples are to be drawn and $\mathbb{X}$ denote its support. If $\mathbb{X}$ is a disconnected set, which is a union of $k$ non-empty disjoint connected open subsets ($\mathbb{X}_i, i = \{1, 2...k\}$), each representing a mode, then $P_X$ is termed as a multimodal distribution with $k$ modes. Following we describe our method to induce multiple modes in the data generating space by engineering the latent space along with inversion. For simplicity, we assume that $P_X$ is a bimodal distribution, albeit all the analysis holds equally well for multimodal distributions.

Let $\mathbf{z}$ denote a latent variable with a distribution $\mathbf{z} \sim P_Z$ and $\mathbb{Z}$ be the support of $P_Z$. Let $g : \mathbb{Z} \to \mathbb{X}$ be a continuous function mapping from $\mathbf{z}$ to $\mathbf{x}$ which serves as an estimator for $P_X$. Let $\mathbb{X}_0 \subseteq \mathbb{X}$ and $\mathbb{X}_1 \subseteq \mathbb{X}$ represent two modes of $P_X$.

**Proposition 1.** Let $\mathbb{Z}_0 \subseteq \mathbb{Z}$ and $\mathbb{Z}_1 \subseteq \mathbb{Z}$ respectively be the inverse images of $\mathbb{X}_0$ and $\mathbb{X}_1$ under $g$, then $\mathbb{X}_0 \cap \mathbb{X}_1 = \Phi$ only if $\mathbb{Z}_0 \cap \mathbb{Z}_1 = \Phi$, where $\Phi$ is an empty set.

*Proof:* In Appendix A.

From proposition 1, to get a bimodal generated distribution, it is necessary to have a bimodal latent distribution, however it is not a sufficient condition. If there exists a mapping $h : \mathbb{X} \to \hat{\mathbb{Y}}$ which maps $\mathbf{x}$, the output of $g$, to another random variable $\hat{\mathbf{y}}$ in such a way that $P_{\hat{Y}}$ is bimodal, one can apply proposition 1 again on $h$ and achieve bimodality on $P_X$. The following corollary to proposition 1 is stated to affirm this fact.

**Corollary 1.1.** Let $\hat{\mathbb{Y}}_0 \subseteq \hat{\mathbb{Y}}$ and $\hat{\mathbb{Y}}_1 \subseteq \hat{\mathbb{Y}}$ be two subsets of $\hat{\mathbb{Y}}$ such that $h : \mathbb{X}_0 \to \hat{\mathbb{Y}}_0$ and $h : \mathbb{X}_1 \to \hat{\mathbb{Y}}_1$ then $\hat{\mathbb{Y}}_0 \cap \hat{\mathbb{Y}}_1 = \Phi$ only if $\mathbb{X}_0 \cap \mathbb{X}_1 = \Phi$. Given $\mathbb{Z}_0 \cap \mathbb{Z}_1 = \Phi$, $\hat{\mathbb{Y}}_0 \cap \hat{\mathbb{Y}}_1 = \Phi$ is a sufficient condition for $\mathbb{X}_0 \cap \mathbb{X}_1 = \Phi$.

Corollary 1.1 asserts if latent distribution is bimodal and $h$ maps $\mathbf{x}$ to any bimodal distribution, the generated distribution ($P_X$) will also be bimodal. However, since both $g$ and $h$ are continuous mappings, conditional generation of samples from $P_X$ (conditioned on modes of $\mathbf{z}$) is possible only if $h$ is in accordance with the modal properties of $\mathbf{z}$. Thus, we introduce a new random variable $\mathbf{y}$ that is an indicator of the modes of $\mathbf{z}$. Formally, $\mathbf{y} := \mathbf{1}_{\mathbf{z} \in \mathbb{Z}_1}$, which implies that $P_Y(\mathbf{y} = 1) = \int_{\mathbb{Z}_1} P_Z \, d\mathbf{z}$

---

[1]Code for implementation can be found at - https://github.com/NEMGAN/NEMGAN

and $P_Y(\mathbf{y} = 0) = \int_{\mathbb{Z}_0} P_Z \; d\mathbf{z}$. With this, we show that if $P_Y$ and $P_{\hat{Y}}$ are close, then the modal properties of $P_X$ will match with those of $P_Z$.

**Proposition 2.** Define a random variable that is an indicator of modes of $P_X$ as follows: $\hat{\mathbf{x}} := \mathbf{1}_{\mathbf{x} \in \mathbb{X}_1}$, $P_{\hat{\mathbb{X}}}(\hat{\mathbf{x}} = 1) = \int_{\mathbb{X}_1} P_X \; d\mathbf{x}$ and $P_{\hat{\mathbb{X}}}(\hat{\mathbf{x}} = 0) = \int_{\mathbb{X}_0} P_X \; d\mathbf{x}$. The minimization of $D_{\mathrm{KL}}(P_{\hat{Y}} || P_Y)$ is equivalent to the minimization of $D_{\mathrm{KL}}(P_{\hat{Y}} || P_{\hat{X}})$.

*Proof:* In appendix A.

**Corollary 2.1.** Minimizing $D_{\mathrm{KL}}(P_{\hat{Y}} || P_{\hat{X}})$ leads to matching of modal properties of $P_{\hat{X}}$ and $P_Y$, that is, $\int_{\mathbb{X}_i} P_X \; d\mathbf{x} = P_Y(\mathbf{y} = i)$, $i \in \{0, 1\}$.

*Proof:* In Appendix A.

Proposition 2 and Corollary 2.1. state an important fact - modal properties of the latent space and the data generating space can be matched by choosing $h$ which would minimize $D_{\mathrm{KL}}(P_{\hat{Y}} || P_Y)$. This also implies that the imbalance (if any) in the modes of the latent distribution is reflected in the data generating distribution. This is especially useful for datasets containing imbalanced classes, which is often the case in real-world applications. The latent space $Z$ can be constructed following the presumed class imbalance ratio in the real data to conditionally generate imbalanced classes.

Using proposition 1 and 2, one can make the data generating distribution to be bimodal, however, produced modes might be degenerated in a sense that $\mathbb{X}_i$'s reduce to singletons (mode collapse). To avoid this degenerative case, we propose to decompose $h$ as a composite of two mappings $h_1 : \mathbb{X} \to \hat{\mathbb{Z}}$ and $h_2 : \hat{\mathbb{Z}} \to \hat{\mathbb{Y}}$. Minimizing a norm distance between the samples of $\mathbb{Z}$ and $\hat{\mathbb{Z}}$ prevents degenerative modes in $P_X$. This is because $h_1$ enforces a unique reconstruction of every sample of $\mathbf{z}$ which in turn ensures that a unique sample of $\mathbf{x}$ is generated by a unique sample of $\mathbf{z}$. The function $h_1$ can be seen as an activity regularizer that would force every unique noise sample within each mode to map to a unique samples in the inversion and the generated spaces.

In the above formulation, however, the generated distribution is not constrained to be close to the distribution of the real data. For that, one has to rely on a technique (such as GANs) that would enforce $g(\mathbf{z})$ to be close to the distribution of the real data. Thus training a GAN according to the proposed formulation, simultaneously enforces the generated data distribution to be close to the real data distribution and match the modal properties of the latent distribution. Conversely, if the latent distribution is chosen in accordance with the modal properties of the real data distribution, the generated data space will be clustered conditioned on the modes of the real data. This is a desirable property because if the modes of the real data signify semantic separation, then by adopting the proposed procedure one can achieve unsupervised semantically meaningful conditional generation. In the following section, we detail the procedure to realize the proposed method in a GAN framework.

## 2.1 REALIZATION USING GENERATIVE ADVERSARIAL NETWORKS

GAN realizes the mapping $g$ using a deep neural network. Additionally, to ensure that $P_X$ is close to the real data distribution a second neural network, the discriminator denoted by $d$, is adversarially trained to differentiate between a true data sample and generated data sample. At the optimum, the generated distribution, $P_X$ approaches the distribution of the real data, $P_{\mathbb{X}_r}$. The inversion network $h = h_2 \circ h_1$ is also modelled using deep neural networks. All neural networks, $g$, $d$ and $h$ are simultaneously trained to optimize the following objective function:

$$\min_{g, h_2 \circ h_1} \max_{d} \left[ \mathbb{E}_{\mathbf{x}_r}[\log d(\mathbf{x}_r)] + \mathbb{E}_{\mathbf{z}}[\log(1 - d \circ g(\mathbf{z})) + ||\mathbf{z} - h_1 \circ g(\mathbf{z})||_p] + D_{\mathrm{KL}}(P_{\hat{Y}} || P_Y) \right] \quad (1)$$

The proposed method can be seen as establishing a chain of continuous functions of random variables - $\mathbf{y} \to \mathbf{z} \to \mathbf{x} \to \hat{\mathbf{z}} \to \hat{\mathbf{y}}$ and imposing a similar modal structures on all of them. The model being completely unsupervised offers several advantages - (a) Once trained, sampling $\mathbf{z}$ from a particular mode confines $\mathbf{x}$ to a unique mode leading to conditional generation, (b) when training by imposing a fixed prior on $\mathbf{z}$, unseen attributes in the data can be discovered, (c) post training, the $h$ network can be used independently to cluster the data, (d) if $\mathbf{z}$ is well-reconstructed by $h_1$, this architecture can be used for inference (Note that this is not guaranteed since the model convergence can be defined only via convergence of $h_2$ even if $\mathbf{z}$ is not reconstructed back by $h_1$), (e)

Mode-collapse when generated $P_X$ becomes independent of the latent random variable $\mathbf{z}$ is avoided in our formulation since $g$ is explicitly forced to produce a multi-modal distribution by the inversion network $h$.

## 3    RELATED WORK

**Data space auto-encoders**

In these class of methods, the idea is to learn a mapping from the data to the latent space and back to the data space. Methods such as vanilla autoencoders (AE) (Hinton & Salakhutdinov (2006), variational autoencoders (VAE) Kingma & Welling (2013), adverarial autoencoders (AAE) Makhzani et al. (2015)), mode regularized GANs (Che et al. (2016)), VAE/GAN (Larsen et al. (2015)) and Conditional VAE (Bao et al. (2017); Sohn et al. (2015)) fall into this category. While these methods offer several advantages, there are two major issues with this idea -(a) without regularization, a naive AE model ends up learning an identity mapping. Further, a vanilla autoencoder cannot be used as a generative model because latent distribution is not tractable and hence cannot be sampled from, (b) although VAE, AAE, VAE/GAN circumvent this problem by using either variational inference or adversarial training, these models cannot find disentangled mappings and thus cannot be used for unsupervised conditional generation tasks. Another approach adopted in Conditional VAEs (Bao et al. (2017); Sohn et al. (2015) is to embed the label information in the latent space along with variational inference setting. However these methods need the label information and it is shown that often the label information is neglected by the encoder Yan et al. (2016).

**Latent space auto-encoders**

In an alternative setting, these class of methods essentially learn mappings from the latent space to the data space and backwards, autoencoding the latent/noise space. These methods have an advantage over the data autoencoders in a sense that it is relatively easy to reconstruct the noise vectors as one has control over the distribution from which they are chosen. Methods such as Adversarially learned inference (ALI) (Dumoulin et al. (2016)), Bidirectional GANs (BiGAN) (Donahue et al. (2016), VEEGAN Srivastava et al. (2017)) and structured GANs (Deng et al. (2017)) are examples of this approach. In ALI and BiGAN, an encoder is learned that would map the data space to the latent space and additionally the discriminator is taken to classify between the tuples of encoded data-real data and latent noise-generated data. However in these methods, the latent space encoding is learned independently of the generated data, in the sense that the real-data is inverted back but not the generated data. This is shown to lead to poor quality reconstruction and possibility of mode collapse in the generated data space Srivastava et al. (2017). VEEGAN on the other hand maps both the real and generated data space to a fixed distribution in order to prevent the problem of mode collapse. While these methods are more advantageous over data-space autoencoders for the task at hand, they still do not learn disentangled relation between the data and the latent space as they invert the data to a semantically-meaningless fixed noise distribution.

**Regularized information maximizers**

Methods such as InfoGAN (Chen et al. (2016)) and CaTGAN (Springenberg (2015)) follow the idea of regularized information maximization (Krause et al. (2010)). These methods train the GAN along with a regularization term to maximize the mutual information between the generated data and parts of the latent space. The methods are shown to learn disentangled mappings between the latent and the data space in a completely unsupervised manner, albeit they might lead to mode collapse because parts of latent space are not reconstructed. Further since there are no explicit function mapping for noise reconstruction, the learned semantic disentanglement may not be completely faithful (Kim & Mnih (2018); Ghosh et al.). Also inference in the data space is not possible because a major part of the latent space is ignored while information maximization.

### 3.1    OUR CONTRIBUTIONS

1. Engineering latent/noise space - Almost all the aforementioned methods use unimodal distribution such as Gaussian or Uniform distribution for the latent space. As we have shown in Proposition 1, multiple modes are produced in the data generating space when the latent space has multiple modes. Further, even if reconstructed perfectly, a unimodal noise cannot be used for conditional generation since the semantic relation between the latent and the

data space is unknown. In this work we propose to engineer the noise space according to the guess one has on the real data distribution. The specifics for the choice of noise space depend upon the task and the guess one has on the data prior.

2. Two-stage latent reconstruction - The latent-space reconstruction network is made a composite of two functions ($h_1$ and $h_2$), one for norm-based reconstruction ($g(h_1(\mathbf{x}_r))$ used in inference) and one to only reproduce the modal properties ($g(\mathbf{z})$ used for conditional generation). Among them, the KL-loss contributes to mode matching and the norm-loss to the variety in the generated data, acting as an activity regularizer.

3. Devoid of approximate/variational inference - In many formulations such as InfoGAN and VEEGAN, the loss involves an entropy term that requires sampling from an unknown (posterior of the latent variable given the real data) distribution which cannot be computed. These methods rely on minimizing a lower bound on the actual cost via variational inference. However, in our case both the loss terms are exact and hence there is no need for constructing bounds.

Based on the above observations, we call our approach Noise Engineered Mode-matching GAN (NEMGAN). During implementation, a multimodal noise is created by adding uniform noise with a multinomial random variable (akin to $\mathbf{y}$) which is selected in accordance to the assumed data prior. The resulting distribution is a multimodal distribution with non-overlapping modes.

## 4 EXPERIMENTS

We conduct three sets of experiments on three real world datasets namely MNIST (LeCun et al. (2010)), CelebA (Liu et al. (2015)) and CIFAR-10 (Krizhevsky & Hinton (2009)), to evaluate our claims. (a) Mode matching - Design noise space in accordance with the semantic imbalance ratio in the data and evaluate conditional generation, (b) Attribute discovery - Impose a fixed but arbitrary prior on noise and see if the generated modes are semantically meaningful, (c) Semantic representation learning - No particular prior is imposed on noise except for the number of modes and evaluate if meaningful representations are learned and, (d) Biased noise engineering - understand the effect of incorrect guess on the priors and explore the possibility to correct the errors in conditional generation using semi-supervised training, (e) Data inference - validate the data reconstruction capabilities of NEMGAN.

### 4.1 MODE MATCHING

We take two semantically 'distinct' classes (digits 0 and 4) for MNIST based on their t-SNE plots (Maaten & Hinton (2008)), albeit this paradigm is applicable to any combination of digits. Three imbalance ratios of 50:50, 25:75, and 02:98 are considered, signifying no imbalance, slight imbalance and heavy imbalance, respectively in digit 0 and 4. Similar class imbalance is induced in $P_Y$ and hence in the latent space $P_Z$, by construction. Figure 1 depicts the images generated by NEMGAN corresponding to two noise modes. It can be seen that even with an imbalance ratio of 02:98, the network is able to conditionally generate semantically separated images. Further to illustrate the mode matching, we plot the class-wise histograms of the distances between the individual samples and their corresponding means in Figure 2, for both the noise and the generated data space. It can be seen that there is evidence for mode matching between latent and generated spaces. As another level of sanity check, we evaluate the classification accuracy of the reconstruction network on the real data without imbalance $h(\mathbf{x}_r)$ and observed accuracy of about 99% for first two and 90% for the case with a class imbalance of 02:98 in the training data. Next to evaluate the semantic separation, we consider images corresponding to digits 3 and 5, which have a significant overlap in the t-SNE, with an imbalance ratio of 30:70. The left pane of Figure 3 shows the conditional generation for the considered classes (3 and 5) which shows no overlap between the classes in spite of their closeness in the data manifold. In fact, the inversion network $h(\mathbf{x}_r)$ achieves a classification accuracy of 98.2 % on real test samples of the digits 3 and 5 in these settings.

We conduct the mode matching experiments with complex color data sets - CIFAR-10 and CelebA. CIFAR in particular is known to have very high intra-class variability. First, we consider two relatively distinct classes namely airplane and frogs, with an imbalance ratio of 30:70. Once again, the modes are matching leading to conditional generation results shown in right pane of Figure 3 with

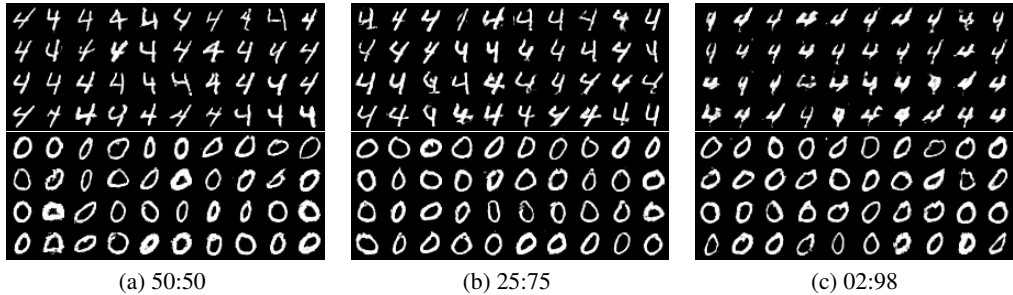

Figure 1: Sample images generated in the experiments with different class ratios. Surprisingly the network is able to do conditional generation even with an imbalance of 02:98 with variety in class corresponding to 2 % (Digit 4).

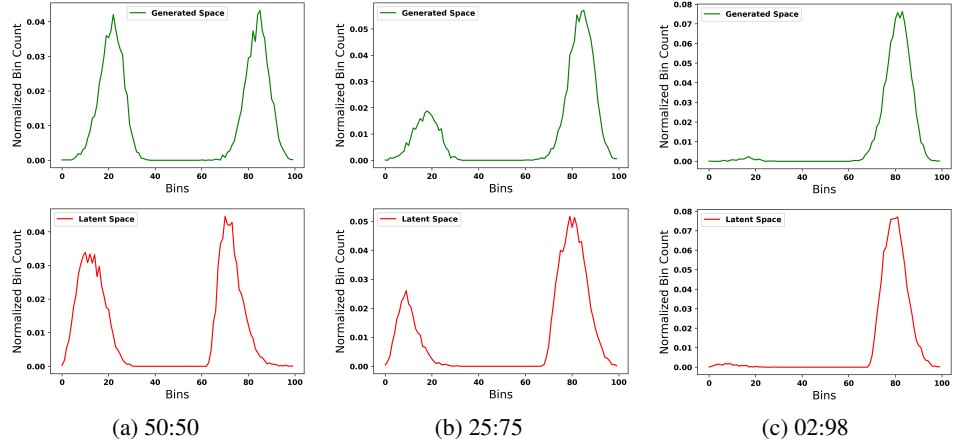

Figure 2: Distribution of sample distances from the corresponding class means for different ratios. The plots indicate mode matching between the latent and generated data space.

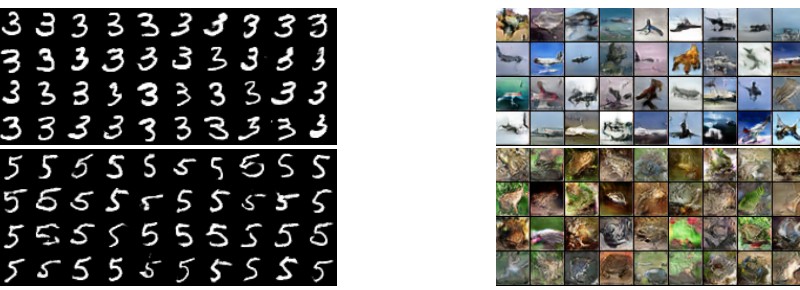

Figure 3: Sample images generated in the experiments with class ratio of 30:70 for digits 3 and 5 in MNIST (left pane), Airplanes and Frogs from CIFAR dataset (right pane). Conditional generation is observed with respective clustering accuracies of 98 and 81.05%.

an inversion network classification accuracy of 81.05% on unseen real data. Similarly, the facial images generated conditioned on presence and absence of glasses, when the latent space is induced with an imbalance of 07:93, shown in first two panes of Figure 4. Furthermore, for an imbalance of 05:95 the network learns to latch on to another facial attribute namely presence or absence of rosy cheeks as observed in last two panes of Figure 4. Another interesting observation in all these experiments is that there is considerable amount of visual variety in the images from each class suggesting reduction of mode collapse, even within a single mode with 2% occurrence. These experiments thus demonstrate the ability of NEMGAN to perform an unsupervised generation conditioned on semantically meaningful classes through mode matching.

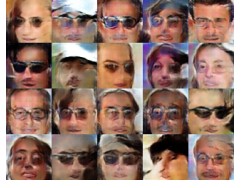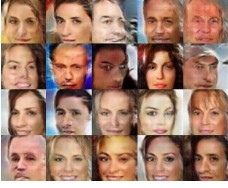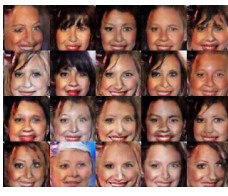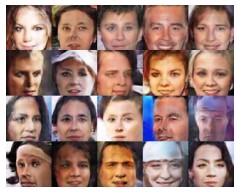

| Glasses | Without glasses | Rosy cheeks | Non-rosy cheeks |

Figure 4: Sample images generated in the experiments with class ratio of 07:93 for glass and non-glass faces, 05:95 for rosy cheeks and non rosy cheeks for the CelebA dataset.

Table 1: Modes captured by different models for the stacked MNIST dataset.

| Method | Modes(Max 1000) |
|---|---|
| DCGAN | 99 |
| VEEGAN | 150 |
| NEMGAN(Ours) | 1000 |

To end the section on mode matching, we evaluate our method on standard mode-counting tasks. Specifically, as mentioned in Srivastava et al. (2017) we used stacked MNIST dataset which is obtained by stacking three random MNIST digits along color channels of an RGB image to obtain color images. This dataset can have 1000 modes corresponding to every possible combination of digit triplets. We used experimental settings identical to Srivastava et al. (2017). Table 1 lists the results of our method and three other methods on discovering the modes in the stacked MNIST. For our method, latent space is a discrete uniform distribution with 1000 modes. It can be seen that NEMGAN is able to discover all 1000 modes, which is quite significant. Appendix D shows generated images and also depicts the outcome of NEMGAN on a eight component GMM arranged over a circle in which case NEMGAN discovers all the modes.

## 4.2 ATTRIBUTE DISCOVERY

In a few real-life scenarios, the class imbalance ratio is unknown. In such cases, an unsupervised technique should discover semantically plausible regions in the data space. To evaluate NEMGANs ability to perform such a task, we perform experiments where sample from $P_Y$ are drawn with an assumed class ratio rather than a known ratio. Two experiments are performed on CelebA, first with the assumption of 2 classes having a ratio of 70:30 and the second with the assumption of 3 classes having a ratio of 10:30:60. In the first experiment, the network discovers visibility of teeth as an attribute to the faces whereas in the second it learns to differentiate between the facial pose angles. Conditional generation from both the experiments are shown in figure 5 and 6, respectively. Note that these attributes are not labelled in the dataset but are discovered by NEMGAN.

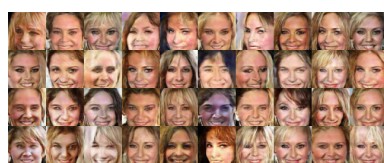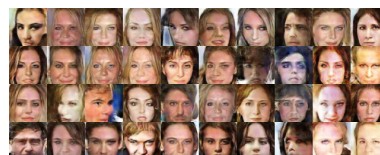

Figure 5: Discovery of the facial attribute smile with teeth visible. Sample images generated in the experiments with class ratio of 70:30 for faces from the CelebA dataset.

## 4.3 SEMANTIC REPRESENTATION LEARNING

In these experiments, we use a $P_Z$ that has $k$ uniform modes and train NEMGAN on MNIST and CIFAR-10. In the first experiment, $k$ is set to 10 for both the datasets based on the fact that there are 10 uniformly distributed semantically distinct classes in both of them. Figure 7 shows the generated MNIST samples. It can be seen that every generated mode corresponds to a digit type. Further a

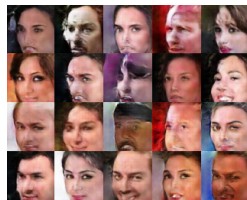 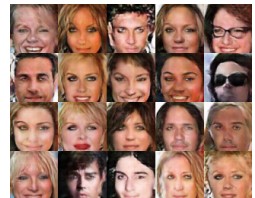 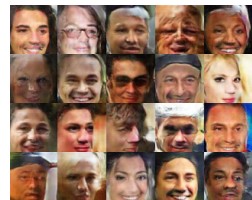

Figure 6: Discovery of the attribute facial pose-angle. Sample images generated in the experiments with class ratio of 10:30:60 for from the CelebA dataset.

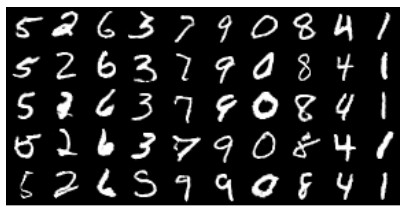 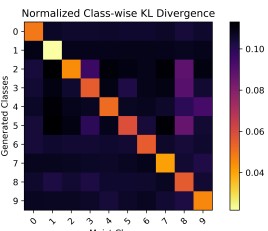

Figure 7: Conditional generation of ten MNIST classes using NEMGAN - left pane: Generated digits, right pane: Class-wise KL between generated and real data.

Table 2: FID values for the MNIST digits generated by different models

| Method | FID score |
|---|---|
| InfoGAN | $26.46 \pm 0.16$ |
| VEEGAN | $203.05 \pm 0.78$ |
| NEMGAN(Ours) | $4.37 \pm 0.13$ |

Table 3: FID values for the CIFAR images generated by different models

| Method | FID score |
|---|---|
| DCGAN | 40 |
| ACGAN | 116.2 |
| NEMGAN(Ours) | 28.73 |

classification accuracy of $96.09\%$ was obtained on the real test data from $h(\mathbf{x}_r)$. To quantify the separability, we compute the class-wise KL divergence between the real and the generated digits (for each class) using a nearest neighbor density estimation technique (Duda et al. (2012)), shown in Figure 7. It can be seen that each mode (class) in the generated data has a unique digit with which the KL is least, signifying a digit-wise semantic separation. To quantify the goodness of the generated data, we compute "Frchet Inception Distance" (FID) which is a measure proposed to capture the closeness of the generated images to real data Heusel et al. (2017). Table 2 lists the FID obtained after 5 runs for NEMGAN and two comparable models, VEEGAN and InfoGAN. It can been seen that NEMGAN achieves the least FID that is close to the real MNIST digits signifying the goodness of the images generated.

Next we repeat the experiment with $k = 10$ on CIFAR-10. Given the huge intra and inter class variety in the CIFAR it is difficult for a fully unsupervised method to generate ten semantically separated modes. However, as seen in the left pane of Figure 8, NEMGAN is surprisingly able to generate ten modes with every mode dominated by one CIFAR class. The generated space seperation is also corroborated from the class-wise KL plot in the right pane of Figure 8. To the best of our knowledge, ours is the first study to report unsupervised class-conditional generation on CIFAR images. Similar to MNIST experiment, NEMGAN performs the best in terms of FID for CIFAR also, as shown in Table 3. Further we conducted the CIFAR experiment with $k = 5$ (5 modes) taking five semantically 'distinct' classes from CIFAR (horses, airplanes, dogs, frogs and automobiles) and the results are shown in Appendix F along with the KL divergence. The class-wise separation is qualitatively more in this case as expected since the classes are 'more' distinct.

## 4.4 ABLATION STUDIES

Our model involves two reconstruction stages for the latent space ($h_1$ and $h_2$). In this section, to study the effect of individual components of the model, we perform the following ablation studies - (a) training a NEMGAN without the $h_1$ network, (b) training a NEMGAN without the $h_2$ network,

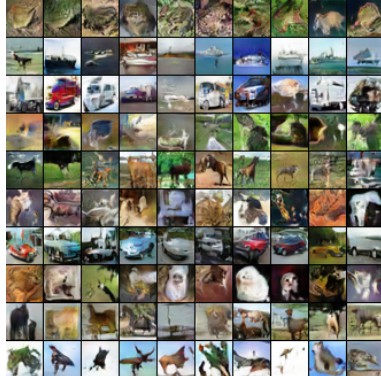 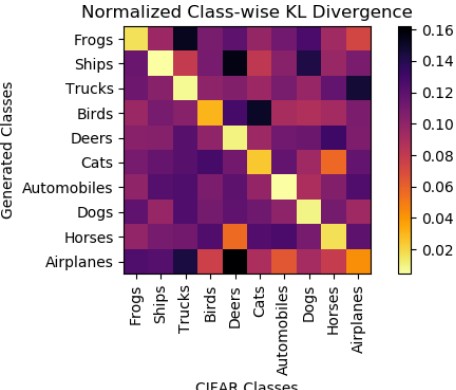

Figure 8: Conditional generation of CIFAR classes using NEMGAN. Every mode in the generated space (one row) is dominated by one CIFAR class in the left pane. The separation is seen evident from the class-wise KL divergence(second pane).

(c) training a conventional GAN with noise engineering. Experiment (a) and (b) are conducted on the MNIST dataset and the output images are shown in Fig 9. It can be seen that the absence of the KL-loss ($h_2$) results in the mixing of classes within each mode and the absence of norm-loss ($h_1$) results in lack of variety within each mode. For example, 1's with serifs are not generated in absence of $h_1$. Results for the experiment (c) is depicted in the Appendix E which suggests that a conventional GAN with latent space engineering cannot separate out the modes. These experiments suggests that the inclusion of the norm-based reconstruction term encourages the model to avoid the intra-class mode collapse unlike in the case of supervised conditional GANs (Rui Shu (2017)).

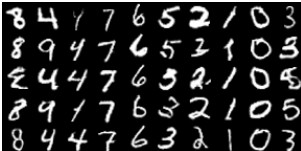 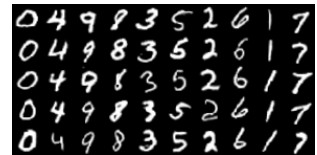

Figure 9: Ablation studies for the latent-space reconstruction terms. Left pane - NEMGAN output in the absence of $h_2$, right pane - NEMGAN output in the absence of $h_1$. It can be seen that absence of $h_2$ results in mode mixing and absence of $h_1$ results in less-variety within modes.

### 4.5 EFFECT OF BIASED NOISE ENGINEERING

In this section we investigate the effect of incorrect guess on the priors for conditional generation task. Samples of two MNIST digits 3 and 5 are taken with an imbalance ratio of 30:70. The imbalance ratio in latent space is induced with errors varying from 5% to 20%. It is seen that NEM-GAN is unable to perform semantic separation of the generated space when trained in unsupervised manner, as shown in Fig. 10. To improve the performance, we explore the possibility of using semi-supervised training, where, 5% of the total training samples (taken without replacement) from each class are used to train the inversion network $h(\mathbf{x}_r)$ (not the generator and discriminator) after every 100 steps of unsupervised training. As shown in Fig. 10, semi-supervised training results in an accurate semantic separation of the generated space. For quantitative evaluation, KL divergence between the generated and real data from the two classes is measured. As shown in Fig. 11, with little supervision, the conditional generation is robust and KL divergence is consistently better than unsupervised training.

### 4.6 INFERENCE

In the final set of experiments, though not the main objective of the paper, we qualitatively evaluate NEMGAN on an inference task where the real images are reconstructed using the outputs of the norm ($L_1$, used in this work) loss layer. Figure 12 depicts the inference for arbitrarily chosen 10 digits from MNIST and CIFAR samples from the two class (Frog Vs. Airplane) experiment (more

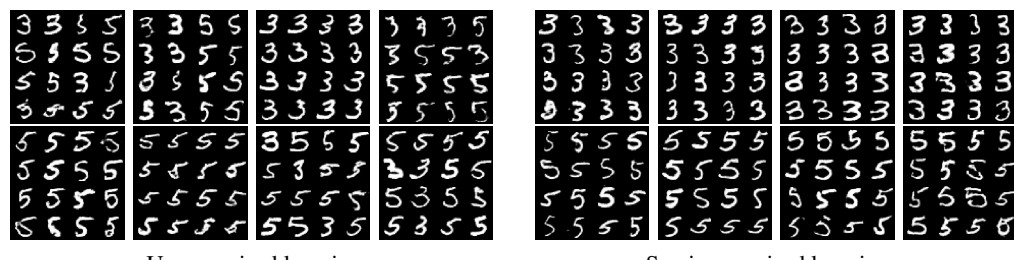

| Unsupervised learning | Semi-supervised learning |

Figure 10: Conditional data generation using NEMGAN with incorrect latent space bias with unsupervised and semi-supervised training. From left-to-right, the error in the latent bias is increased from 5% to 20%. While completely unsupervised training does not separate out the semantic modes, but 5% supervision does.

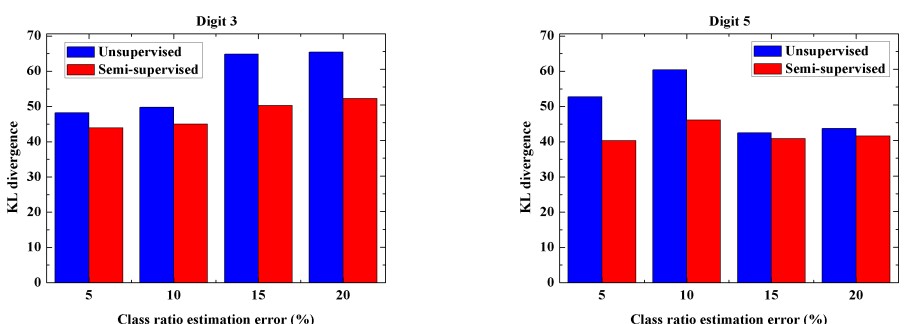

Figure 11: Evaluation of conditional generation using KL divergence in presence of errors in the latent prior, with unsupervised and semi-supervised training. It is seen that with little supervision the conditional generation is robust and KL divergence is consistently better than unsupervised training.

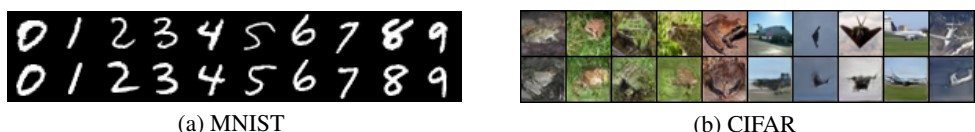

| (a) MNIST | (b) CIFAR |

Figure 12: Inference results of MNIST and CIFAR data. Samples from real test data(top row) are passed through the $g(h_1(\mathbf{x}_r))$ network to generate the images(bottom row).

images with qualitative comparison is given in the Appendix). We have observed that the semantic category is almost always preserved. Given that the network was trained with a skewed class ratio of 70-30, it is significant that a class-wise faithful reconstruction is obtained.

# 5 CONCLUSION

We introduced a method for unsupervised conditional generation using generative adversarial networks through engineered latent space inversion. We demonstrated through multiple experiments, that certain desirable properties could be induced in the generated data by creating a latent noise space that is engineered in accordance to the need. Being fully unsupervised, NEMGAN can be used in multiple applications spanning from conditional generation to data space clustering. It also has several desirable side effects like reduction of the infamous mode collapse. Even though NEMGAN is fully unsupervised, we believe its performance can be improved by augmenting it with some supervision.

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

## A. Proofs for Propositions

**Proposition 1:** Let $\mathbb{Z}_0 \subseteq \mathbb{Z}$ and $\mathbb{Z}_1 \subseteq \mathbb{Z}$ respectively be the inverse images of $\mathbb{X}_0$ and $\mathbb{X}_1$ under $g$, then $\mathbb{X}_0 \cap \mathbb{X}_1 = \Phi$ only if $\mathbb{Z}_0 \cap \mathbb{Z}_1 = \Phi$, where $\Phi$ is an empty set.

*Proof:* Given $\mathbb{X}_0 \cap \mathbb{X}_1 = \Phi$, assume, $\mathbb{Z}_0 \cap \mathbb{Z}_1 \neq \Phi \implies \exists \mathbf{z}_i \in \mathbb{Z}_0 \cap \mathbb{Z}_1$. Given $\mathbf{z}_i \in \mathbb{Z}_0$, let $g(\mathbf{z}_i) = \mathbf{x}_{i0} \in \mathbb{X}_0$ and similarly, given $\mathbf{z}_i \in \mathbb{Z}_1$, $g(\mathbf{z}_i) = \mathbf{x}_{i1} \in \mathbb{X}_1$. Since $g$ is a continuous function, $\mathbf{x}_{i0} = \mathbf{x}_{i1} = \mathbf{x}_i \implies \mathbf{x}_i \in \mathbb{X}_0 \cap \mathbb{X}_1$ contradicting the fact that $\mathbb{X}_0 \cap \mathbb{X}_1 = \Phi$, hence $\mathbb{Z}_0 \cap \mathbb{Z}_1 = \Phi$.

**Proposition 2:** Define a random variable that is an indicator of modes of $P_X$ as follows: $\hat{\mathbf{x}} := \mathbf{1}_{\mathbf{x} \in \mathbb{X}_1}$, $P_{\hat{\mathbb{X}}}(\hat{\mathbf{x}} = 1) = \int_{\mathbb{X}_1} P_X \, d\mathbf{x}$ and $P_{\hat{\mathbb{X}}}(\hat{\mathbf{x}} = 0) = \int_{\mathbb{X}_0} P_X \, d\mathbf{x}$. The minimization of $D_{\mathrm{KL}}(P_{\hat{Y}} || P_Y)$ is equivalent to the minimization of $D_{\mathrm{KL}}(P_{\hat{Y}} || P_{\hat{X}})$.

*Proof:* As we know

$$D_{\mathrm{KL}}(P_{\hat{Y}} || P_Y) = \sum_{\hat{\mathbf{y}}} P_{\hat{Y}} \log \frac{P_{\hat{Y}}}{P_Y} \tag{2}$$

$$= \sum_{\hat{\mathbf{y}}} \left( P_{\hat{Y}} \log P_{\hat{Y}} - P_{\hat{Y}} \log P_Y \right) \tag{3}$$

$$= \sum_{\hat{\mathbf{y}}} P_{\hat{Y}} \log P_{\hat{Y}} - P_{\hat{Y}}(\hat{\mathbf{y}} = 0) \log P_Y(\mathbf{y} = 0) - P_{\hat{Y}}(\hat{\mathbf{y}} = 1) \log P_Y(\mathbf{y} = 1) \tag{4}$$

$$= \sum_{\hat{\mathbf{y}}} P_{\hat{Y}} \log P_{\hat{Y}} - P_{\hat{Y}}(\hat{\mathbf{y}} = 0) \log \int_{\mathbb{Z}_0} P_Z \, d\mathbf{z} - P_{\hat{Y}}(\hat{\mathbf{y}} = 1) \log \int_{\mathbb{Z}_1} P_Z \, d\mathbf{z} \tag{5}$$

Since $\int_{\mathbb{Z}_i} P_Z \, d\mathbf{z} = \int_{\mathbb{X}_i} P_X \, d\mathbf{x}$, equation 3 can be written as

$$D_{\mathrm{KL}}(P_{\hat{Y}} || P_Y) = \sum_{\hat{\mathbf{y}}} P_{\hat{Y}} \log P_{\hat{Y}} - P_{\hat{Y}}(\hat{\mathbf{y}} = 0) \log \int_{\mathbb{X}_0} P_X \, d\mathbf{x} - P_{\hat{Y}}(\hat{\mathbf{y}} = 1) \log \int_{\mathbb{X}_1} P_X \, d\mathbf{x} \tag{6}$$

Since $\int_{\mathbb{X}_i} P_X \, d\mathbf{x} = P_{\hat{X}}(\hat{\mathbf{x}} = i)$, by definition, equation 6 can be written as

$$D_{\mathrm{KL}}(P_{\hat{Y}} || P_Y) = \sum_{\hat{\mathbf{y}}} P_{\hat{Y}} \log P_{\hat{Y}} - P_{\hat{Y}}(\hat{\mathbf{y}} = 0) \log P_{\hat{X}}(\hat{\mathbf{x}} = 0) - P_{\hat{Y}}(\hat{\mathbf{y}} = 1) \log P_{\hat{X}}(\hat{\mathbf{x}} = 1) \tag{7}$$

$$= \sum_{\hat{\mathbf{y}}} \left( P_{\hat{Y}} \log P_{\hat{Y}} - P_{\hat{Y}} \log P_{\hat{X}} \right) \tag{8}$$

$$= D_{\mathrm{KL}}(P_{\hat{Y}} || P_{\hat{X}}) \tag{9}$$

**Corollary 2.1:** Minimizing $D_{\mathrm{KL}}(P_{\hat{Y}} || P_{\hat{X}})$ leads to matching of modal properties of $P_{\hat{X}}$ and $P_Y$, that is, $\int_{\mathbb{X}_i} P_X \, d\mathbf{x} = P_Y(\mathbf{y} = i)$, $i \in \{0, 1\}$.

*Proof:* The optimum value of $D_{\mathrm{KL}}(P_{\hat{Y}} || P_{\hat{X}})$ is achieved when $P_{\hat{X}}(\hat{\mathbf{x}} = i) = P_Y(\mathbf{y} = i)$. Further from equation 9, $P_Y(\mathbf{y} = i) = P_{\hat{Y}}(\hat{\mathbf{y}} = i)$. Hence, $P_{\hat{X}}(\hat{\mathbf{x}} = i) = P_Y(\mathbf{y} = i)$, $i \in \{0, 1\}$.

## B.1. Qualitative Comparison with existing architectures

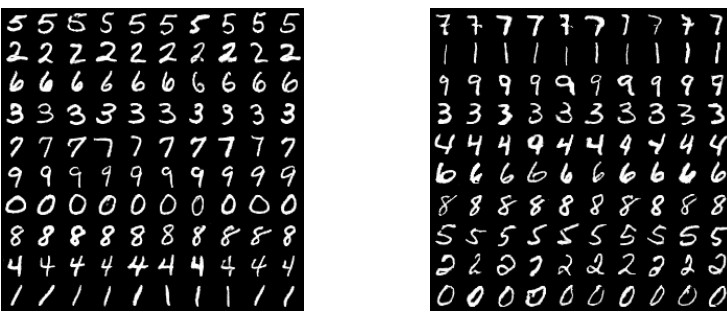

(a) Conditional generation using NEMGAN. (b) Conditional generation using InfoGAN.

Figure 13: Conditional generation comparison.

## B.2. Inference comparison

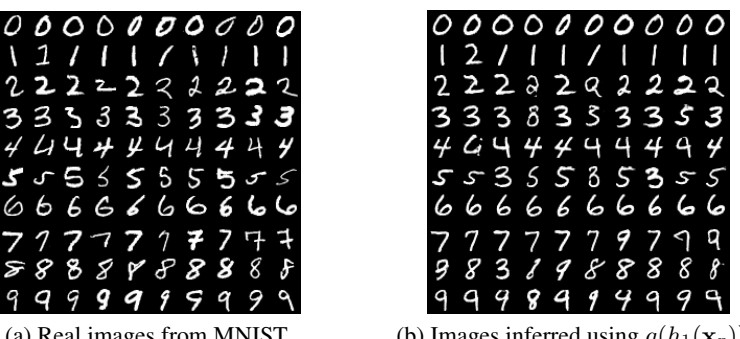

(a) Real images from MNIST      (b) Images inferred using $g(h_1(\mathbf{x}_r))$.

Figure 14: Inference results of MNIST data. Samples from real test data are passed through the $g(h_1(\mathbf{x}_r))$ network to generate the images.

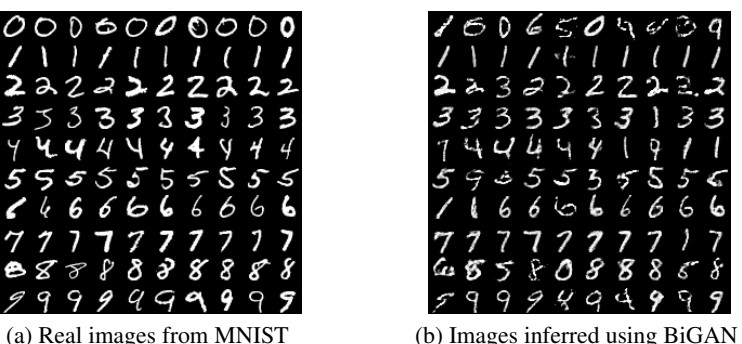

(a) Real images from MNIST      (b) Images inferred using BiGAN

Figure 15: Inference results of MNIST data from BiGAN.

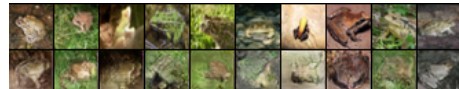

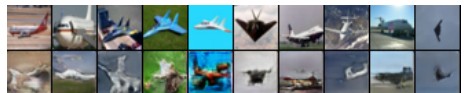

(a) Top: real images of frogs from CIFAR dataset, bottom: Inference results obtained using NEMGAN.

(b) Top: real images of airplane from CIFAR dataset, bottom: Inference results obtained using NEMGAN.

Figure 16: Top: real images from CIFAR, bottom: Inference results obtained using NEMGAN.

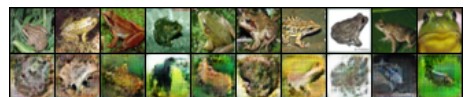

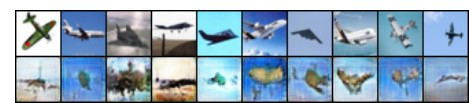

(a) Top: real images of frogs from CIFAR dataset, bottom: Inference results obtained using ALI.

(b) Top: real images of airplane from CIFAR dataset, bottom: Inference results obtained using ALI.

Figure 17: Top: real images from CIFAR dataset, bottom: Inference results obtained using ALI.

## C. Mode Separation

In this work, semantics of the data refer to the modes in data distribution. These semantics represent different attributes of the samples and are separated out by the proposed method. For a better understanding, experiments are conducted with samples of only a single digit type from the MNIST dataset. Samples of digit 7 and 4 are considered for this purpose. NEMGAN is trained with a discrete uniform latent space with 10 modes and the generated images are shown in Fig. 18. Each row in Fig. 18 corresponds on one latent space mode and shows different attributes of the considered digits. For example, the fifth row in left pane contains generated images of digit 7 with slits. Similarly in right pane, the third row contains images of digit 4 with a closed notch. Note that, even with images of a single digit, no mode collapse is observed with NEMGAN.

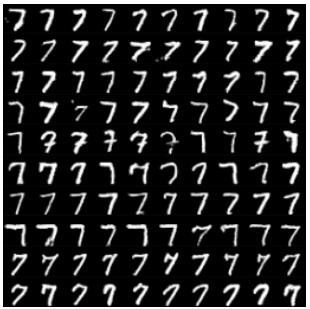

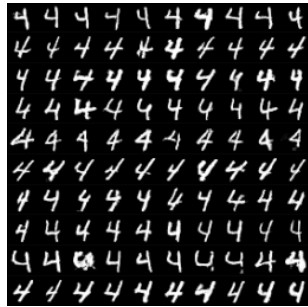

Figure 18: Demonstration of mode separation using NEMGAN. Every row in each figure depicts sample from a mode when the NEMGAN is trained only with a single digit type with a latent space with ten modes.

## D. Mode counting using NEMGAN

We trained NEMGAN for mode counting experiment on stacked MNIST dataset. It is able to generate 993 modes. Some of the generated images are shown in Fig. 19. Similar performance is observed in 8 component GMM experiment, as shown in Fig. 21.

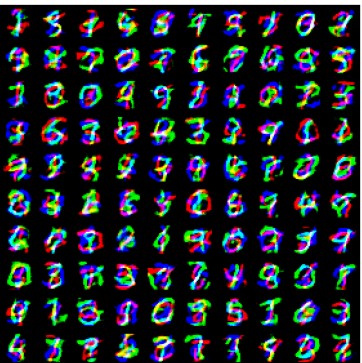

Figure 19: Mode counting experiment result for stacked MNIST dataset. NEMGAN is able to produce variety of modes after training.

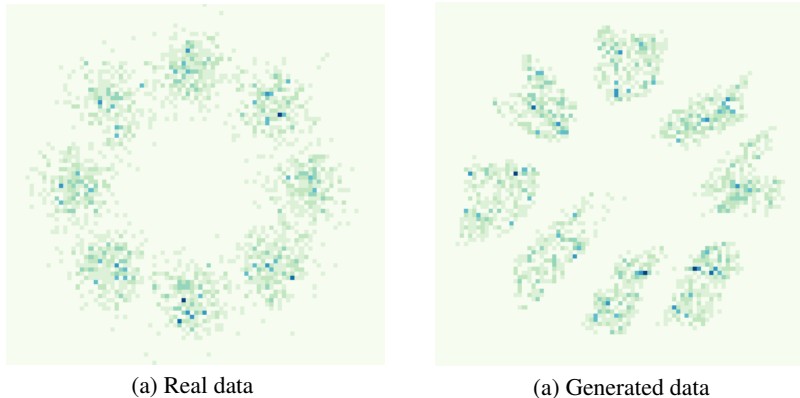

(a) Real data                                    (a) Generated data

Figure 20: Density plots of true data and NEMGAN generator output for 8 component GMM arranged over a circle

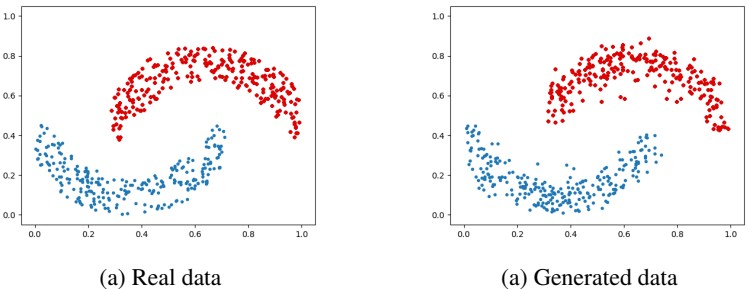

(a) Real data                                    (a) Generated data

Figure 21: Density plots of true data and NEMGAN generator output for two classes arranged in an interleaved crescent pattern.

## E. Conventional GAN with latent space engineering

Proposition 1 and 2 show that if there exists a mapping from the generated data space back to the latent space, the generated space is deemed to have as many modes as in the latent space. Multimodal latent space is necessary to generate date with multiple modes, however, it is not sufficient. To understand this, the mode matching experiment performed with images of digits 3 and 5 in section 4.1 is repeated with conventional GAN. Images of digit 3 and 5 with an imbalance ratio of 30:70 are used to train the conventional GAN with a bimodal latent space having a modal mass of 30:70. Observations of the experiment are shown in Fig. 22. The conventional GAN is not able to separate

the modes and most of the samples drawn are from the dominant mode Fig. 22(b). On the other hand, NEMGAN separates out the modes accurately.

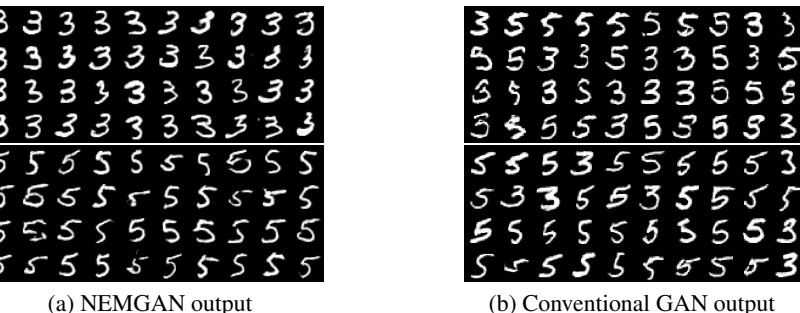

(a) NEMGAN output          (b) Conventional GAN output

Figure 22: NEMGAN and conventional GAN output in presence of latent space engineering. Images of digit 3 and 5 with an imbalance ratio of 30:70 are used in the experiment.

## F. Conditional generation of CIFAR images

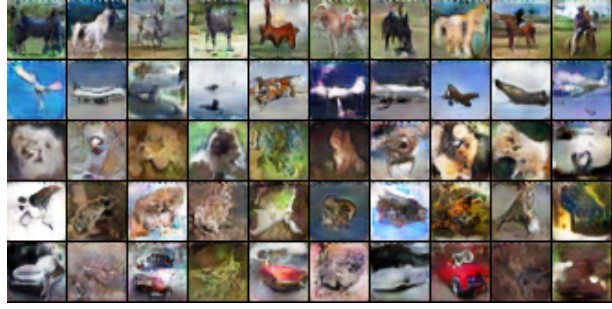

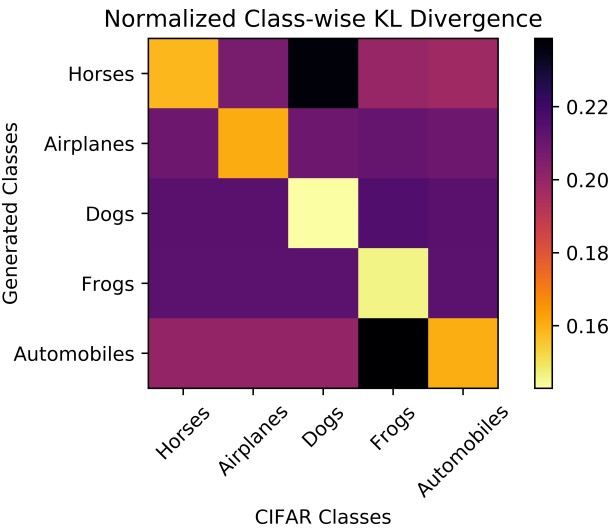

Figure 23: Conditional generation of CIFAR 5 classes using NEMGAN and corresponding KL divergence matrix. It is seen, and affirmed by KL divergence, that every mode in the generated space (one row) represents one CIFAR class.

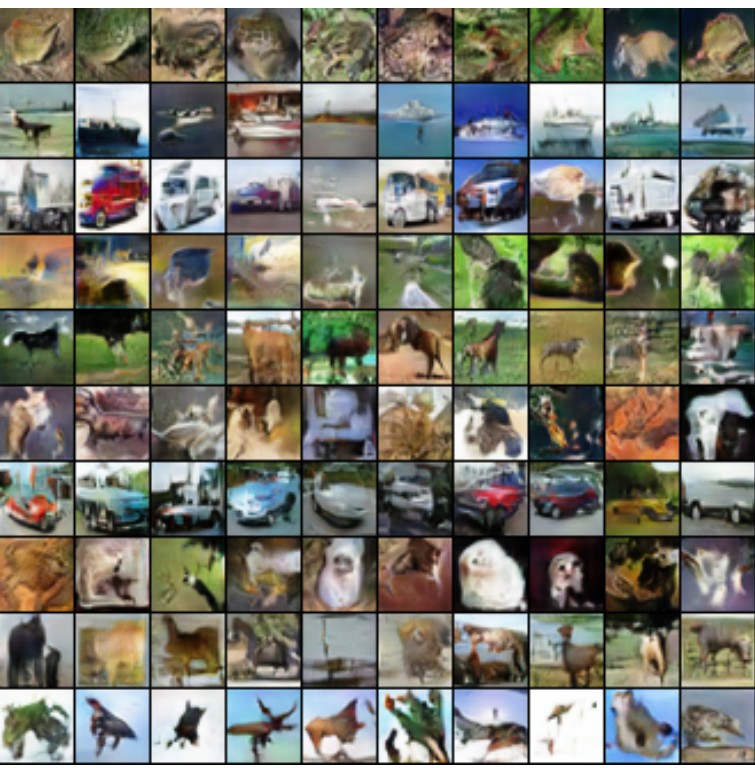

Figure 24: Conditional generation of CIFAR 10 classes using NEMGAN. It is seen that the every mode in the generated space (one row) is dominated by one CIFAR class.

