# OpenReview forum: "Unsupervised Conditional Generation using noise engineered mode matching GAN"
_ICLR.cc/2019/Conference_

### Official Review · AnonReviewer2 · 2018-11-02

**Rating:** 6
**Confidence:** 4

**Review:**

The paper proposes simple modifications to GAN architecture for unsupervised conditional image generation. The authors achieve this by making the distribution of noise z dependent on variable y that can depend on the label distribution when available. This involves learning to predict the input noise z as well as y from the generated image. The qualitative results shown for unsupervised conditional image generations using the approach are convincing.

Pros:
	- The paper is well written and easy to follow.
	- The simple modification to the noise distribution leads to good results on unsupervised conditional image generation.
	- Minimizes loss terms exactly instead of lower bound as is commonly done in other similar unsupervised approaches.
	- Theoretical justifications for the approach are convincing.

Cons:
	- The paper can be strengthened by including ablation studies on the loss term for the z reconstruction.
	- How will InfoGAN and VEEGAN compare with similar loss terms for the z reconstruction added to their objective?
	- It will be useful to show FID and other similar scores to better evaluate the learned generative model. Including mode counting, experiments will strengthen the paper.
	- ACGAN can suffer from the issue of generating images where it is easy to infer y [1]. This leads to mode collapse within each category/class. Will the proposed approach suffer from similar issues?

[1] Shu, Rui, Hung Bui, and Stefano Ermon. "AC-GAN Learns a Biased Distribution."

---

> ### Author Response · Authors · 2018-11-19
> **Response to Reviewer #2**
>
> We thank the reviewer for the constructive feedback and comments. Below are our responses for the concerns raised. We are happy to answer further questions, if any.
>
> Q1. The paper can be strengthened by including ablation studies on the loss term for the z reconstruction.
>
> Ans: Thank you for this suggestion. We have now included the ablation studies in the experimental section. The following are the observations -
>
> Our model involves two reconstruction stages for the latent space ($h_1$ and $h_2$). To study the effect of individual components of the model, we perform the following ablation studies -  (a) training a NEMGAN without the $h_1$ network, (b) training a NEMGAN without the $h_2$ network, (c) training a conventional GAN with noise engineering. Experiment (a) and (b) are conducted on the MNIST dataset and the output images are shown in Fig 9. It can be seen that the absence of the KL-loss ($h_2$) results in the mixing of classes within each mode and the absence of norm-loss ($h_1$) results in lack of variety within each mode. For example, 1's with serifs are not generated in absence of $h_1$. Results for the experiment (c) is depicted in the Appendix E which suggests that a conventional GAN with latent space engineering cannot separate out the modes. These experiments suggests that the inclusion of the norm-based reconstruction term encourages the model to avoid the intra-class mode collapse unlike in the case of supervised conditional GANs [1].
>
> Q2. How will InfoGAN and VEEGAN compare with similar loss terms for the z reconstruction added to their objective?
>
> Ans: The original VEEGAN has a latent space reconstruction term in it by construction. We included the z reconstruction term in the official implementation of the InfoGAN and found that it is unable to produce the MNIST images. We believe that this might be because of the fact that the Q network, which is used to maximize the mutual information, is made a part of the discriminator. Thus, performing a joint task of maximizing mutual information between the categorical code and the generated images as well as reconstructing the noise term is not feasible. That is perhaps the reason authors of InfoGAN terms ‘z’ as the irreducible noise. On the contrary, our architecture has the discriminator and the latent reconstructor decoupled from each other which is the reason our method performs multiple tasks such as conditional generation, data inference etc.
>
> Q3. It will be useful to show FID and other similar scores to better evaluate the learned generative model. Including mode counting, experiments will strengthen the paper.
>
> Ans: Thank you again for this suggestion. We have included the FID values in Table 2 and 3 of the current version of the paper and compared with the existing methods. We have also included the mode counting experiments on standard stacked MNIST data and a toy 8 component GMM data in Appendix D. Mode counting results are summarized in Table 1 of the current version of the paper and it is found that NEMGAN captures the most number of modes.
>
> Q4. ACGAN can suffer from the issue of generating images where it is easy to infer y [1]. This leads to mode collapse within each category/class. Will the proposed approach suffer from similar issues?
>
> Ans: This is precisely the reason we broke down the z-reconstruction loss into two parts - KL and norm-based. The below paragraph (from Sec. 2 of the paper) summarizes the idea -
>
> Using proposition 1 and 2, one can make the data generating distribution to be bimodal, however, produced modes might be degenerated in a sense that $\sX_i$'s  reduce to singletons (mode collapse).  To avoid this degenerative case, we propose to decompose  $h$  as a composite of two mappings $h_1:{\sX}\rightarrow \hat{\sZ}$ and $h_2:\hat{\sZ}\rightarrow \hat{\sY}$. Minimizing a norm distance between the samples of $\sZ$ and $\hat{\sZ}$ prevents degenerative modes in $P_X$. This is because $h_1$ enforces a unique reconstruction of every sample of $\rvz$ which in turn ensures that a unique sample of $\rvx$ is generated by a unique sample of $\rvz$. The function $h_1$ can be seen as an activity regularizer that would force every unique noise sample within each mode to map to a unique samples in the inversion and the generated spaces.
>
> These ideas are reconciled with the ablation experiments (Sec. 4.4) where the inclusion of norm-based loss is seen to avoid the intraclass mode collapse. This point has been brought out in the current version of the paper.
>
> [1] Shu, Rui, Hung Bui, and Stefano Ermon. "AC-GAN Learns a Biased Distribution."

---

### Official Review · AnonReviewer1 · 2018-11-05
**A well-written paper; would be appreciated if the motivation were clearer and the method more principle**

**Rating:** 5
**Confidence:** 3

**Review:**

This paper is concerned with the so-called conditional generation, which was descried as the task of sampling from an unknown distribution conditioned on semantics of the data. This proposed method aims at achieving this by using a latent distribution engineered according to a certain data prior. Experimental results showed that the proposed method seems to produce good results.

I have several questions about the motivation and the method in the paper. First, it is not clear to me how the "semantics" of the data was defined. Is it given by visual inspection? Is it possible to find it with some automated method? Second, the authors seem to advocate the idea that data of a k-mode distribution should be generated from a k-mode latent distribution. It might be useful in certain scenarios; however, it is not clear why the transformation from the latent to the observed does not change the number of modes or why keeping the same number of modes would endow the latent distribution a "semantics" meaning. We know that a k-mode distribution can be obtained by applying a smooth nonlinear transformation to a Gaussian or uniform distribution and, similarly, a k-mode distribution can be transformed to a single-mode distribution with a smooth mapping. So I am not sure why engineering the latent distribution this way can give it a "semantics" meaning. Should we try to enforce a kind of smoothness of the transformation, by, say, penalizing high derivative values? Third, the experimental results seem nice, but the lack of comparisons blurs the advantage of the proposed method. How is the result produced by GAN compared to the reported one? How did the original GAN with the engineering latent distribution work? It would be appreciated if the authors could address these issues more clearly.

---

> ### Author Response · Authors · 2018-11-17
> **Response to Reviewer #1 (part 1 of 2)**
>
> We thank the reviewer for the insightful comments and below are our responses for the concerns raised. We are happy to answer further questions, if any.
>
> Q1.  First, it is not clear to me how the "semantics" of the data was defined. Is it given by visual inspection? Is it possible to find it with some automated method?
>
> Ans: By semantics of the data we refer to the modes in the data distribution. Further, the modes are defined as disjoint connected open subsets in the support of the distribution of the data (Please refer to the opening paragraph of section 2). These modes may correspond to ‘natural’ classes in the data (as in MNIST data where every mode correspond to a digit) or separable attributes in data space (like smile with teeth versus no teeth in the CelebA dataset). It is akin in spirit to the concept of ‘clusters’ in any unsupervised task wherein the idea of a cluster depends on the task at hand in which case the semantics (modes) are implicitly defined.
>
> Our claim is that the properties of the latent space are imposed on the generated data space and hence ‘semantics’ (also called the modes or disjoint connected open subsets) similar to that of the latent space are enforced on the generated space. We do NOT claim that these modes (or semantics) in the generated data are always visually interpretable. However, if the true data space has visually interpretable modes, then so is the generated data space. For example, we have shown experimentally that each of the 10 modes in the generated MNIST dataset correspond to a digit type, without any mixup.
>
> To drive home this point, we have conducted further experiments where we consider only a single digit type from the MNIST dataset and train a NEMGAN with a discrete uniform latent space with 10 modes. It is observed (and reported in the revised version of the paper) that the generated data space has exactly 10 distinct modes. Further, while the data from each mode exhibits some level of separation (for instance when we consider digit 7, one of the modes contains 7s with the vertical line slit), not all of them are visually interpretable. Please refer to the Appendix C of the paper for the figures from this experiment.
>
> In summary, conceptually by semantics we mean the modal properties of the true data space which is obtained (or assumed) by the prior information that one has about the data. The following text has been added to the introduction -
> “The proposed model thus preserves the semantics of the data in the sense that each mode in the generated space correspond to a certain data semantic”.
>
> Q2: The authors seem to advocate the idea that data of a k-mode distribution should be generated from a k-mode latent distribution. It might be useful in certain scenarios; however, it is not clear why the transformation from the latent to the observed does not change the number of modes.  We know that a k-mode distribution can be obtained by applying a smooth nonlinear transformation to a Gaussian or uniform distribution and, similarly, a k-mode distribution can be transformed to a single-mode distribution with a smooth mapping.
>
> Ans:  As pointed out rightly, a unimodal distribution can be transformed into multimodal distribution and vice versa. Ideally, a neural network (the GAN generator), being a non linear transformation is supposed to transform a unimodal latent distribution to a multimodal distribution but it has been found practically that this often does not happen. It is a well-known fact that a GAN tends to collapse the modes in the observed space even though the true data space has multiple modes.
>
> Also, if the latent space is a union of k non-empty disjoint connected open subsets, no continuous function could map it to a set with more than k disjoint subsets (Proposition 1). Based on this, our aim in this work is to propose a method which enforces a GAN to produce a distribution with multiple modes (disjoint open subsets) and thus avoiding mode collapse. We achieve so by training an inversion network which maps the observed data space back to a multimodal latent space, in tandem with a regular GAN, by using Proposition 1 twice  (once on the latent to data mapping and once on the data to latent mapping). It is noteworthy that in all  our experiments the latent space is designed to be a union of k non-empty disjoint connected open subsets.
>
> In summary, owing to the fact that neural networks are deterministic continuous functions, we have shown (in Proposition 1 and 2) that if there exists a mapping from the observed data space back to the latent space, the observed space is deemed to have as many modes as in the latent space. It is this inversion (with the KL-loss) which ensures that the transformation from the latent to the observed does not change the number of modes.

---

> ### Author Response · Authors · 2018-11-17
> **Response to Reviewer #1 (part 2 of 2)**
>
> Q3. Why keeping the same number of modes would endow the latent distribution a "semantics" meaning. So I am not sure why engineering the latent distribution this way can give it a "semantics" meaning. Should we try to enforce a kind of smoothness of the transformation, by, say, penalizing high derivative values?
>
> Ans: As discussed while answering the first question, semantics refer to the modes in the true data. Latent space engineering alone wouldn’t endow semantics in the observed space but training an inversion network in tandem would, in the sense that the number (and the properties) of modes in the observed space will be same as that in the latent space (which is assumed to be same as the true data space).
>
> While enforcing a smooth transformation on the generator might produce a multi modal distribution, it is not guaranteed. However, a multimodal latent space in tandem with an inversion mapping from the data back to a multimodal distribution does so (Please refer to Proposition 1 and 2).
>
> Q4. The experimental results seem nice, but the lack of comparisons blurs the advantage of the proposed method. How is the result produced by GAN compared to the reported one? How did the original GAN with the engineering latent distribution work? It would be appreciated if the authors could address these issues more clearly.
>
> Ans: Thank you for this suggestion. We have added new experiments with a conventional GAN trained with the engineered latent space. Please refer to Appendix E (Figure 22) for details. The below are the observations.
>
> We took 2 MNIST digits (3 and 5) with a skew of 30:70 and trained a conventional GAN with a bimodal latent space with a modal mass of 30:70. It was observed that the conventional GAN could not separate the modes and most of the samples drawn are from the dominant mode. On the contrary, it is shown in the paper that our model can separate out the modes.
>
> Mode counting results for stacked MNIST dataset and FID values are included in Table 1, 2 and 3 of the current version of the paper, respectively.

---

### Official Review · AnonReviewer3 · 2018-11-06
**Modification to GAN construction which induces bias to encourage mode matching between latent and data space.**

**Rating:** 5
**Confidence:** 3

**Review:**

This paper presents a GAN construction which encourages the latent space to mode-match the data distribution.  This allows for unsupervised class-label inference (experimentally verified for two tasks).

I think the paper is of low significance, but the approach outlined is interesting.

Unfortunately, I think the work is slightly let down by the presentation (there are many typos, and the first couple of sections could do with a rewrite), as well as a lack of rigorous experimentation.  I believe that the paper is also missing references to the conditional VAE literature, which shares many similarities (at least in application) with the described approach.

Pros:
- Some theoretical justification for the approach taken.
- Early evidence that the method allows for latent space class separation, given a prior on number of classes.

Cons:
- A little more experimental evidence would be welcome.  E.g. why is the result for CIFAR 10 not shown---hard to understand how helpful the inductive bias is for a general problem.
- No discussion of conditional VAEs (which were designed with a very similar goal in mind).
- No discussion of why decomposing h in the manner in which they did was appropriate.
- Would be nice to see a more detailed study of how adding supervision + varying the strength of the inductive bias affects performance.

---

> ### Author Response · Authors · 2018-11-25
> **Response to Reviewer #3 (part 1 of 2)**
>
> Thank you for the insightful reviews. Your review helped in improving our paper significantly. Below are the point-wise responses for the concerns raised.
>
> Q1. I think the paper is of low significance, but the approach outlined is interesting.
>
> Ans: Thank you for noting that it is an interesting approach. We acknowledge that the significance of the paper was not brought out correctly in the previous version of the paper. However, we believe that this work is of considerable significance because of the following reasons.
>
> 1. This is one of the very few methods that could do unsupervised class-conditional generation in spite of severe class-imbalance.
>
> 2. There are a lot of practical scenarios, for example healthcare data, where the classes are severely imbalanced with unknown class-labels with known skew ratio. The proposed method becomes extremely useful in such cases to automate/aid the annotation process and augment the data from the less-occurring class.
>
> 3. We have shown that this method can be potentially used to discover the unknown hidden semantic attributes in the data.
>
> 4. A simple engineered latent-space inversion network with KL-term enables multiple tasks such as clustering, conditional generation, mode recovery and inference (Please note that different methods are proposed for individual tasks while ours can do multiple tasks at once with a simple modification).
>
> 5. Theory-wise, to the best of our knowledge, this is the first work that proposes to tweak the latent space and reconstruct it such that the generated data space inherits the properties of the latent space.
>
> 6. To the best of our knowledge, this is the first work that attempts an unsupervised conditional generation on CIFAR.
>
> Q2. Unfortunately, I think the work is slightly let down by the presentation (there are many typos, and the first couple of sections could do with a rewrite),
>
> Ans: We acknowledge this fact and we have now thoroughly reworked on the introduction section and corrected all the typos.
>
> Q3. …… as well as a lack of rigorous experimentation
>
> Ans: Following the guidelines of the reviewers, we have conducted  more experiments to strengthen the paper. The following is a list of new experiments that are included in the revised version of the paper.
>
> 1. Quantification of the results by computing state-of-the-art metrics and comparison with the existing methods.
>
> 2. Ablation studies to ascertain the effect of reconstruction-loss terms.
>
> 3. Mode matching/counting experiments on standard tasks such as GMM rings and stacked MNIST.
>
> 4. Quantification of strength of inductive-bias with semi-supervision.
>
> Q4. I believe that the paper is also missing references to the conditional VAE literature, which shares many similarities (at least in application) with the described approach.
>
> Ans:- Thank you for pointing out the mistake. We have now discussed about these in the revised paper.
>
> Q5. - A little more experimental evidence would be welcome.  E.g. why is the result for CIFAR 10 not shown---hard to understand how helpful the inductive bias is for a general problem.
>
> Ans:-  Thank you for the suggestions. We have included several new experiments and results.
>
> CIFAR is understood to be a very difficult dataset to perform unsupervised tasks, since the classes are not well separated. Every class could be tied to another semantically ‘close’ class. For example, the ‘deer’ class is very close to the ‘horse’ class, the ‘automobile’ class is very close to the ‘truck’ class etc. We claim that the modes in data space matches to the latent space, albeit the modes in the data space do not necessarily correspond to classes, especially if the classes are not well-separated. That is the reason we showed the results only on 5-classes of CIFAR which are semantically separated and demonstrated that they are recovered.
>
> Having said that, your comments made us further experiment with 10 class CIFAR and we could observe that class-wise modes are separated with our architecture yielding better FID than a vanilla GAN. We have reported this result in the current version of the paper.

---

> ### Author Response · Authors · 2018-11-25
> **Response to Reviewer #3 (part 2 of 2)**
>
> Q6. - No discussion of conditional VAEs (which were designed with a very similar goal in mind).
>
> Ans:- Thank you again for pointing this out. We have included discussions on relevant papers in section 3 of the revised version of the paper.
>
> Q7. - No discussion of why decomposing h in the manner in which they did was appropriate.
>
> Ans: We have addressed this question in Section 2 of the paper in the following paragraph.
>
> Using proposition 1 and 2, one can make the data generating distribution to be bimodal, however, produced modes might be degenerated in a sense that $\sX_i$'s  reduce to singletons (mode collapse).  To avoid this degenerative case, we propose to decompose  $h$  as a composite of two mappings $h_1:{\sX}\rightarrow \hat{\sZ}$ and $h_2:\hat{\sZ}\rightarrow \hat{\sY}$. Minimizing a norm distance between the samples of $\sZ$ and $\hat{\sZ}$ prevents degenerative modes in $P_X$. This is because $h_1$ enforces a unique reconstruction of every sample of $\rvz$ which in turn ensures that a unique sample of $\rvx$ is generated by a unique sample of $\rvz$. The function $h_1$ can be seen as an activity regularizer that would force every unique noise sample within each mode to map to unique samples in the inversion and the generated spaces.
>
> The intuitive explanation for having a norm-based reconstruction term is that it avoids the intra-mode collapse. Further to ascertain this fact, we have conducted ablation studies retaining only either of the terms and included the results in the revised version.
>
> Q8: Would be nice to see a more detailed study of how adding supervision + varying the strength of the inductive bias affects performance.
>
> Ans: Thanks for this suggestion. This helped us to understand our method in a deeper way. We conduct following experiments to quantify the effect of strength of inductive bias.
>
> 1. Induce an incorrect bias on the latent space and quantify the extent of mode separation.
>
> 2. Induce an incorrect bias and incorporate supervision while training the inversion network (as suggested) and study the extent of mode separation.
>
> It is found out that the method can perform well despite errors in the inductive bias with a little supervision during training. Please refer to Section 4.5 for details.

---

### Author Response · Authors · 2018-11-17
**Thanks for the insightful reviews.**

We are thankful to the reviewers for the constructive feedback and comments. It has helped us in improving the quality of the paper and understanding the work better. We shall address all the concerns of the reviewers one by one through suggested experiments and revisions.

---

### Author Response · Authors · 2018-11-26
**Fully revised paper uploaded.**

Dear Reviewers,

We thank you all for the thoughtful reviews that helped us revise the paper substantially.

We have tried to incorporate all the suggestions given by the reviewers and the revised paper is uploaded in the portal.

Thank you again.

Authors.

---

### Meta-Review · Area_Chair1 · 2018-12-15
**Use of multimodal prior for mode conditional generation but lacks convincing experiments to demonstrate its use/benefits**

**Confidence:** 4
**Recommendation:** Reject

**Metareview:**

The paper uses a multimodal prior in GANs and reconstructs the latents back from images in two stages to match the generated data modes to the latent space modes. It is empirically shown that this can prevent mode collapse to some extent (including intra-class collapse). However the paper lacks a comparison with state of the art GANs that have been shown to get better FID scores (~21 for SN-GAN [1] vs ~28 in the paper) so the benefit here is unclear, particularly in cases when the mode prior is unknown. Similarly for other applications used in the paper such as inference and attribute discovery, it falls short of demonstrating quantitative improvements with the approach. For example, there is a growing body of work on unsupervised disentanglement in generative models with several metrics to measure it, which could be used to evaluate the attribute discovery performance. R1 has brought up the point of lack of comparisons which the AC agrees with. Authors have made revisions in the paper including some comparisons but these feel insufficient to establish the benefits of the method over state of the art in preventing mode collapse.

A borderline paper as reflected in the reviewer scores but can be made stronger with experiments showing convincing improvements over state of the art in at least one of the applications considered in the paper.


[1] Miyato, T., Kataoka, T., Koyama, M., & Yoshida, Y. (2018). Spectral normalization for generative adversarial networks. ArXiv Preprint ArXiv:1802.05957.